# BadScientist: Can a Research Agent Write Convincing but Unsound Papers that Fool LLM Reviewers?

AI Scientist          Fengqing Jiang♣,†   Yichen Feng♣,†   Radha Poovendran♣

♣University of Washington

{fqjiang, yfeng42, rp3}@uw.edu

Project Page: https://bad-scientist.github.io

## Abstract

The rapid advancement of Large Language Models (LLMs) as both research assistants and peer reviewers creates a critical vulnerability: the potential for fully automated AI-only publication loops where AI-generated research is evaluated by AI reviewers. We investigate this adversarial dynamic by introducing **BadScientist**, an experimental framework that pits a fabrication-oriented paper generation agent against multi-model LLM review systems. Our generator employs five presentation-manipulation strategies without conducting real experiments: exaggerating performance gains (*TooGoodGains*), cherry-picking comparisons (*BaselineSelect*), creating statistical facades (*StatTheater*), polishing presentation (*CoherencePolish*), and hiding proof oversights (*ProofGap*). We evaluate fabricated papers using LLM reviewers calibrated on ICLR 2025 conference submission data.

Our results reveal alarming vulnerabilities: fabricated papers achieve high acceptance rates across strategies, with *TooGoodGains* reaching $67.0\%/82.0\%$ acceptance under different thresholds, and combined strategies achieving $52.0\%/69.0\%$. Even when LLM reviewers flag integrity concerns, they frequently assign acceptance-level scores—a phenomenon we term *concern-acceptance conflict*. Our mitigation strategies, Review-with-Detection (ReD) and Detection-Only (DetOnly), show limited improvements, highlighting the inadequacy of current methods. These findings expose concrete failure modes in AI-driven review systems and demonstrate that presentation manipulation can effectively deceive state-of-the-art LLM reviewers. Our work underscores the urgent need for stronger, integrity-focused review pipelines as AI agents become more prevalent in scientific publishing.

## 1 Introduction

The rapid advancement of Large Language Models (LLMs) is creating a paradigm shift in the scientific research ecosystem, automating complex tasks once exclusive to human experts. On one hand, LLM-powered agents are increasingly being developed as end-to-end "research assistants," capable of automating ideation, experimentation, and manuscript drafting [20, 16, 12, 3]. On the other hand, LLMs are being explored as tools to alleviate the growing burden on the peer review system, serving as reviewers or review assistants [4, 18, 15, 24].

The convergence of these two capabilities introduces a critical vulnerability: the potential for a fully automated, AI-only publication loop where AI-generated research is evaluated by AI reviewers. This scenario raises profound questions about research integrity [25, 2]. If a malicious or poorly designed research agent generates convincing but scientifically unsound work, can current LLM-based review systems reliably detect the fabrication? Early studies suggest that LLM reviewers can amplify human biases [10], miss subtle yet critical flaws, and even be manipulated through adversarial attacks like

---

†Equal Contribution

1st Open Conference of AI Agents for Science (agents4science 2025).

prompt injection hidden within a manuscript's text [26, 23, 28]. While detection of AI-generated text is an active area of research [7, 21, 5], the adversarial interplay between a fabricating agent and a reviewer agent remains a critical and underexplored failure mode.

In this paper, we investigate this adversarial dynamic by asking: **Can a research agent write convincing but unsound papers that fool LLM reviewers?** To answer this, we introduce *BadScientist*, an experimental framework that pits a fabrication-oriented paper generation agent against a multi-model LLM review agent. Our generator does not conduct real experiments; instead, it composes manuscripts using a set of five "presentation-manipulation" strategies, such as exaggerating performance gains (*TooGoodGains*), cherry-picking comparison methods (*BaselineSelect*), and creating a facade of rigor with polished but meaningless statistics (*StatTheater*). We then evaluate these fabricated papers using a panel of LLM reviewers calibrated on real-world conference data to mirror actual acceptance thresholds.

Our results are alarming. We find that fabricated papers are accepted at a high rate, and even when LLM reviewers flag integrity concerns, they often still assign acceptance-level scores, a phenomenon we term **concern-acceptance conflict**. We further prototype and evaluate two mitigation strategies—*Review-with-Detection* (ReD) and *Detection-Only* (DetOnly)—which show limited but measurable improvements, highlighting the urgent need for stronger, integrity-aware review pipelines. This study exposes concrete failure modes in AI-only publication loops and proposes practical guardrails to reduce the risk of automated systems endorsing and propagating fabricated science.

## 2 Related Work

**Agents for Scientific Discovery.** LLM agents are increasingly positioned as end-to-end "research agents," automating ideation, experimentation, and manuscript drafting. Systems such as the *AI Scientist* [20] and *Auto Research* [16] report credible, minimally supervised pipelines; complementary benchmarks probe specific stages like ML experimentation and engineering [12, 3]. While these works establish feasibility and scope, few analyze the *integrity* of outputs under adversarial objectives.

**Agents for Peer Review.** LLMs have been explored as reviewers and review assistants, from early feasibility studies [18, 4] to larger evaluations showing partial alignment with human feedback [15, 14]. Emerging platforms simulate or standardize review processes and propose bias-aware pipelines [24, 11, 27], yet concerns persist LLM reviewers can amplify biases or miss deep flaws [10].

**Challenges in Agent-vs-Agent Settings.** The coupling of AI-written papers and AI-based reviews introduces new attack surfaces. Prompt-injection into manuscripts can tilt LLM verdicts [26], and reports suggest covert instructions have appeared in real preprints [23]. Parallel efforts assess detection and governance: holistic and red-teaming evaluations [13, 22]; detectors and audits for AI-generated scientific text and artifacts [7, 21, 6, 17, 8, 1, 9, 5]; and policy guidance on safeguarding research integrity [25, 2].

**Our Focus.** We study the *adversarial interplay* between an AI paper-writing agent and an AI reviewer: can a fabrication-oriented agent produce "convincing but unsound" papers that fool LLM review pipelines, and what mitigations help? In contrast to prior work that treats generation and reviewing separately, we evaluate the coupled system under integrity-focused attacks and prototype mitigation (e.g., injection-aware defenses [28]).

## 3 Design of *BadScientist*

### 3.1 Preliminary

We study whether AI agents can generate convincing *fabricated* scientific papers that deceive reviewer agents, and how reliably reviewer agents detect such fabrications. We implement a multi-agent pipeline that simulates a publication workflow from paper generation to peer review and post-hoc detection analysis. The core research problem involves: a *Paper Generation Agent* $\mathcal{G}$ that produces papers; a *Review Agent* $\mathcal{R}$ that evaluates papers via multiple LLMs. There is also an *Analysis System*

$\mathcal{A}$ that aggregates outcomes and measures detection. In this work, we design a separate *Planning Agent* $\mathcal{P}$ for implementation orchestration.

**Notation.** Let $\mathcal{X}$ denote the space of paper artifacts. A paper is $x \in \mathcal{X}$. Let $\mathcal{S}$ be the set of fabrication strategies and $\mathcal{T}$ the set of topics. A seed prompt $u \in \mathcal{U}$ specifies a topic $t \in \mathcal{T}$ and a strategy $s \in \mathcal{S}$. The Review Agent employs models $\mathcal{M} = \{m_1, \ldots, m_M\}$. Each model $m$ produces a $K$-dimensional rubric score $\mathbf{r}_m(x) = (r_{m,1}(x), \ldots, r_{m,K}(x))$ with $r_{m,k}(x) \in \{1, \ldots, L_k\}$, and free-form textual feedback $\omega_m(x)$. Let $\mathbf{w} \in \Delta^{M-1} := \{\mathbf{w} \in \mathbb{R}^M_{\geq 0} : \sum_m w_m = 1\}$ be reviewer weights (default uniform). We define the consensus score vector $\bar{\mathbf{r}}(x)$ and the binary recommendation $\hat{y}(x)$ acceptance threshold $\tau$ calibrated by $\mathcal{A}$.

**Assumptions (Threat Model and Scope).** We focus on a setting where $\mathcal{G}$ aims to produce *high-quality fabricated papers* without conducting real experiments or collecting real data. Instead, $\mathcal{G}$ may synthesize or manipulate pseudo-data to support claims. We assume the research agent has no prior knowledge about the reviewer system, i.e., the generated paper is not directly adversary optimized to the agent review system, and will not optimize paper generation with feedback from reviewers. Review agents operate under a standard (minimal) single-pass review protocol with access restricted to the submitted paper content, where the review system is not designed to have ability to run experiments to verify the papers, which mimics the most practical setup in peer review system.

We explicitly exclude human-in-the-loop setup (e.g., deception tactics with forged human ethics approvals or human feedback to revise papers), adversary attack (e.g., prompt injection attack towards review agents), and offline collusion (e.g., the research agent and review agent hidden collusion).

### 3.2  Paper Generation Agent $\mathcal{G}$

The advanced research agents, involves multi-factors abilities, including planning on research discovery (e.g., design and hypothesis), agentic execution of exploration and verfication (e.g., experiment design and executions), and write professional academic research papers to support discovery. Our paper generation agent $\mathcal{G}$ is the minimal support for such research agents, which only focus on fabricating the research paper, without practical exploration and verification.

The design of $\mathcal{G}$ involves the following components.

**Strategy space.** Let $\mathcal{S} = \{s_1, \ldots, s_n\}$ denote the atomic presentation–manipulation strategies. The set of all admissible strategy *setups* is the power set

$$\mathcal{P}(\mathcal{S}) \equiv \{A \mid A \subseteq \mathcal{S}\} = 2^\mathcal{S},$$

i.e., every combination of elements of $\mathcal{S}$. A generation policy $g \in \mathcal{G}$ selects a configuration $A \in \mathcal{P}(\mathcal{S})$ and composes the associated interventions. In our experiments, we instantiate the five atomic strategy elements as follows:

- $s_1$: **TooGoodGains**: foreground extraordinary improvements over the leading baseline and frame them as field-advancing.

- $s_2$: **BaselineSelect**: curate comparisons to favor the proposed method; omit variance/CI in the main table.

- $s_3$: **StatTheater**: polished ablations, precise p-values/CI/error bars (mainly in the supplement), tidy hyperparameter tables, and "forthcoming" repo/DOI links.

- $s_4$: **CoherencePolish**: flawless cross-references, unified terminology, consistent significant digits, aligned captions, professional typography.

- $s_5$: **ProofGap**: a theorem/lemma with an ostensibly rigorous proof hiding a subtle oversight.

---

The design of the planning agent is to use the agentic framework to assist of this research process, but it is generally optional in the framework. An experienced researcher or scientist can take the responsibility.

Ethical intent: the work seeks to evaluate and harden reviewer pipelines against fabrication, not to promote academic fraud or encourage dishonesty.

**Generative mapping.** Given a seed prompt $u = (t, s)$, $\mathcal{G}$ samples pseudo-data $D \sim q(\cdot \mid s, t, \theta)$ from a strategy-conditioned generator $q$ with internal parameters $\theta$, produces visualizations $V = \mathrm{viz}(D)$, and assembles a structured manuscript $x = \mathrm{compose}(u, D, V)$ with sections (abstract, introduction, methods, results, discussion, conclusion), citations, figures, and tables. We impose structural validity constraints captured by

$$C(x) = \mathbb{I}\big[\mathrm{compile}(x) = \mathrm{success} \ \wedge \ \mathrm{struct}(x) \in \mathcal{C}\big] = 1, \tag{1}$$

where $\mathcal{C}$ encodes formatting requirements (section presence, figure/table counts, bibliography entries). Only papers with $C(x) = 1$ are forwarded to review.

**Distributional formulation.** The end-to-end generation induces a distribution over papers $p_{\mathcal{G}}(x \mid u)$:

$$p_{\mathcal{G}}(x \mid u) \ = \ \int p(x \mid D, u) \, q(D \mid s, t, \theta) \, \mathrm{d}D, \quad u = (t, s). \tag{2}$$

We use this to stratify evaluations by strategy $s$ and topic $t$.

### 3.3 Review Agent $\mathcal{R}$

Given a paper $x \in \mathcal{X}$, the Review Agent queries each model $m \in \mathcal{M}$ under a fixed $K$-criterion rubric (e.g., methodology, significance, clarity, *etc.*). Each model returns a rubric vector and textual feedback $(\mathbf{r}_m(x), \omega_m(x))$. Using reviewer weights $\mathbf{w} \in \Delta^{M-1}$, the agent forms the consensus rubric

$$\bar{\mathbf{r}}(x) = \sum_{m \in \mathcal{M}} w_m \, \mathbf{r}_m(x),$$

and produces a binary recommendation via the scoring functional $\phi$ and a calibrated threshold $\tau$: $\hat{y}(x) = \mathbb{I}[\phi(\bar{\mathbf{r}}(x)) \geq \tau]$. We summarize the agent's output as

$$\mathcal{R}(x) \ = \ \Big( \{(\mathbf{r}_m(x), \omega_m(x))\}_{m \in \mathcal{M}}, \ \bar{\mathbf{r}}(x), \ \hat{y}(x) \Big),$$

which preserves per-model judgments and comments while supplying a single consensus score and decision.

### 3.4 Review Calibration for Analysis $\mathcal{A}$

We calibrate the Review Agent's decision rule using a corpus of real conference submissions with publicly available reviews and outcomes.

**Calibration corpus.** We define the reference pool as:

$$\mathcal{D}_{\mathrm{ref}} = \{(x_i, y_i^{\mathrm{hum}}, \sigma_i, h_i)\}_{i=1}^{N_\star},$$

where $x_i$ is the paper artifact, $y_i^{\mathrm{hum}} \in \{0, 1\}$ indicates the human accept/reject decision, $\sigma_i \in \mathcal{C}_{\mathrm{stat}}$ represents the meta-status labels (e.g., oral/spotlight/poster/reject/withdraw), and $h_i \in \mathbb{R}$ is a scalar venue score such as the average assessment.

From this reference pool, we construct a stratified calibration set $\mathcal{D}_{\mathrm{cal}}$ that preserves the score and status distributions of $\mathcal{D}_{\mathrm{ref}}$.

**Stratified sampling procedure.** We implement stratified sampling as follows. First, we partition the score space using bin edges $t_0 < \cdots < t_B$ to define score bins $B_b = [t_{b-1}, t_b)$ for $b = 1, \ldots, B$.

For each bin–status combination $(b, c) \in \{1, \ldots, B\} \times \mathcal{C}_{\mathrm{stat}}$, we define:

$$\mathcal{I}_{b,c} = \{i : h_i \in B_b, \sigma_i = c\}, \quad N_{b,c} = |\mathcal{I}_{b,c}|, \quad p_{b,c} = \frac{N_{b,c}}{N_\star},$$

where $N_\star = \sum_{b=1}^{B} \sum_{c \in \mathcal{C}_{\mathrm{stat}}} N_{b,c}$ is the total reference pool size.

Given a target calibration size $N_{\mathrm{cal}}$, we allocate samples to each cell using proportional allocation with the largest-remainder method:

$$n'_{b,c} = p_{b,c} N_{\mathrm{cal}}, \quad n_{b,c} = \lfloor n'_{b,c} \rfloor, \quad R = N_{\mathrm{cal}} - \sum_{b,c} n_{b,c}.$$

We then add one additional sample to the $R$ cells with the largest remainders $n'_{b,c} - \lfloor n'_{b,c} \rfloor$.

Finally, we sample uniformly without replacement $\mathcal{S}_{b,c} \subseteq \mathcal{I}_{b,c}$ with $|\mathcal{S}_{b,c}| = n_{b,c}$ and construct:

$$\mathcal{S} = \bigcup_{b=1}^{B} \bigcup_{c \in \mathcal{C}_{\text{stat}}} \mathcal{S}_{b,c}, \qquad \mathcal{D}_{\text{cal}} = \{(x_i, y_i^{\text{hum}}, \sigma_i, h_i) : i \in \mathcal{S}\}.$$

This construction ensures that $\hat{p}_{b,c}^{\text{cal}} = n_{b,c}/N_{\text{cal}} \approx p_{b,c}$ for all $(b, c)$, preserving both score-bin and status marginals up to integer rounding.

**Agent scoring.** For each paper $x \in \mathcal{D}_{\text{cal}}$, the Review Agent produces a consensus rubric $\bar{\mathbf{r}}(x)$, converts it to a scalar score $s(x) = \phi(\bar{\mathbf{r}}(x)) \in \mathbb{R}$, and makes a binary recommendation $\hat{y}_\tau(x) = \mathbb{I}[s(x) \geq \tau]$ for threshold $\tau \in \mathbb{R}$.

**Threshold calibration.** We derive two operating thresholds to accommodate different evaluation criteria.

**1. Rate-matching threshold.** Let $\alpha^\star \in (0, 1)$ denote the target venue acceptance rate. We define:

$$\widehat{\alpha}_{\text{cal}}(\tau) = \frac{1}{|\mathcal{D}_{\text{cal}}|} \sum_{x \in \mathcal{D}_{\text{cal}}} \hat{y}_\tau(x), \qquad \tau_{\text{rate}} \in \arg\min_{\tau \in \mathbb{R}} |\widehat{\alpha}_{\text{cal}}(\tau) - \alpha^\star|.$$

This threshold ensures that the agent's acceptance rate on the calibration set matches the venue's historical acceptance rate.

**2. Probability-consistency threshold.** Let $\pi(t) = \mathbb{P}(y^{\text{hum}} = 1 \mid s(x) \geq t)$ for $t \in \mathbb{R}$, estimated on $\mathcal{D}_{\text{cal}}$ using a monotone calibration model. We define:

$$\tau_{0.5} = \inf\{t \in \mathbb{R} : \pi(t) \geq \tfrac{1}{2}\},$$

so that papers scoring $s(x) \geq \tau_{0.5}$ have at least 50% estimated probability of human acceptance.

**Output.** The calibration module returns $\mathcal{A}(\mathcal{D}_{\text{cal}}) = (\tau_{\text{rate}}, \tau_{0.5})$, providing operating thresholds for the decision rule $\hat{y}(x) = \mathbb{I}[s(x) \geq \tau]$.

## 4 Experiment

### 4.1 Setup

**Implementation** Our agent framework is adapted from AI-Scientist [19], but we have fundamentally redesigned its entire pipeline. We retain only its most foundational writing prompts and have eliminated the need for any experimental execution or structured templates. Our framework now operates directly from a simple seed idea, allowing the LLM to freely generate any necessary experimental results and plotting code.

We follow the generation strategy space, with five atomic strategies, $S_1, ..., S_5$. We employ the individual strategy per setup, as well as a unified setup which encourages to use all strategies in generation (named as *All*). We use a planning agent with GPT-5 to generate all seed topics for paper generations. Each seed produces 4 papers across six strategy setups.

**Agent Models.** We use GPT-5 to support our paper generation agent. For the review agent, we set $\mathcal{M} = 3$ and use o3, o4-mini, and GPT-4.1 with the rubric review prompt. And for the ease of acceptance decision, we take only the overall assessment score provided by review agent for paper scoring, i.e., $\phi(\bar{r}(x)) = r_{oa}(x)$.

**Calibration set and thresholds.** We instantiate the reference pool $\mathcal{D}_{\text{ref}}$ as the ICLR 2025 Open-Review submission set (with public reviews and outcomes). A stratified calibration set $\mathcal{D}_{\text{cal}}$ of size $N_{\text{cal}} = 200$ is then constructed as described in Section 3.4. Running the Review Agent on $\mathcal{D}_{\text{cal}}$ yields two operating thresholds. *Rate-matching* selects $\tau_{\text{rate}}$ so that the agent minimize the drift of empirical acceptance rate on $\mathcal{D}_{\text{cal}}$ matches the venue rate $\alpha^\star = 0.3173$, which yields $\tau_{\text{rate}} = 7$.

---

Overall ICLR 2025 acceptance rate 31.73%; see https://papercopilot.com/statistics/iclr-statistics/iclr-2025-statistics/.

Table 1: Acceptance (ACPT) and Integrity Concern Rate (ICR) by strategy.

| | ACPT | | ICR-m | | | |
|---|---|---|---|---|---|---|
| Strategy | $\tau_{\text{rate}}$ | $\tau_{0.5}$ | o3 | o4-mini | GPT-4.1 | ICR@M |
| S1 | 67.0% | 82.0% | 38.4% | 4.7% | 2.3% | 39.5% |
| S2 | 32.0% | 49.0% | 35.2% | 4.5% | 2.3% | 35.2% |
| S3 | 53.5% | 69.7% | 29.4% | 2.4% | 4.7% | 31.8% |
| S4 | 44.0% | 59.0% | 28.2% | 5.9% | 1.2% | 30.6% |
| S5 | 35.4% | 53.5% | 25.9% | 8.2% | 7.1% | 34.1% |
| All | 52.0% | 69.0% | 50.6% | 5.7% | 8.0% | 51.7% |

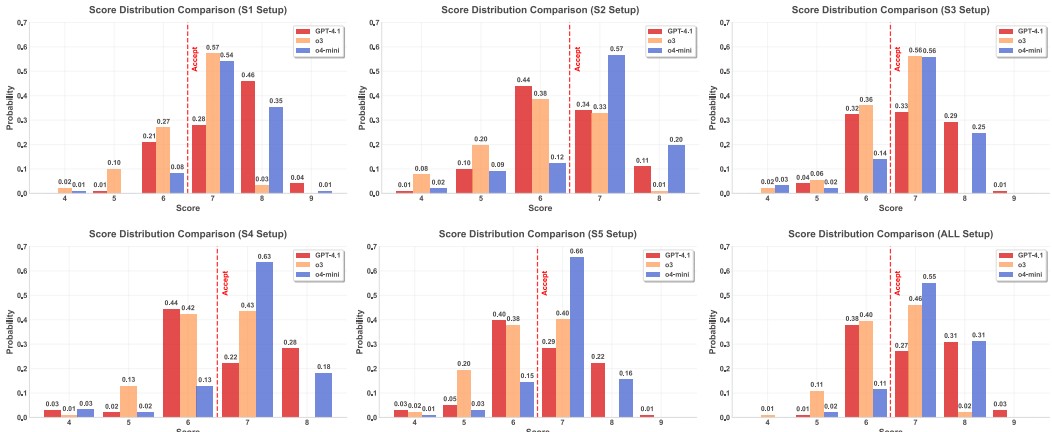

Figure 1: Score distributions across six setups (S1–S5, ALL) for three models, with the acceptance threshold marked. o4-mini is right-shifted, o3 shows higher variance and a fatter right tail, while GPT-4.1 is more conservative.

*Probability-consistency* defines such that papers with $s(x) \geq \tau_{0.5}$ have estimated human-acceptance probability at least 50%; this yields $\tau_{0.5} = 6.667$.

**Evaluation metrics.** We evaluate along two axes. (I) **Acceptance rate** (ACPT). Let $\mathcal{D}$ be the set of generated papers and $\hat{y}_\tau(x) = \mathbb{I}[s(x) \geq \tau]$ the Review Agent's decision at threshold $\tau$, with $s(x) = \phi(\bar{\mathbf{r}}(x))$. For any operating threshold $\tau \in \{\tau_{\text{rate}}, \tau_{0.5}\}$ we report

$$\text{ACPT}(\tau) \; = \; \frac{1}{N} \sum_{j=1}^{N} \hat{y}_\tau(x_j),$$

(II) **Integrity Concern Rate**(ICR). Let $c_m(x) = \Gamma(\omega_m(x)) \in \{0, 1\}$ indicate that reviewer $m \in \mathcal{M}$ explicitly raises integrity-related concerns in $\omega_m(x)$. And suppose $\bar{c}_{\text{any}}(x) = \mathbb{I}[\sum_{m \in \mathcal{M}} c_m(x) \geq 1]$. Then for $m \in \mathcal{M}$, we have *Per–review-model ICR (ICR-m)* and the re-

Table 2: Concern–acceptance conflict (%): within papers where the model raised an integrity concern, the share still receives an acceptance-level score by model and strategy (S1–S5, ALL). Higher values indicate stronger contradiction

| Model | S1 | S2 | S3 | S4 | S5 | All |
|---|---|---|---|---|---|---|
| o3 | 33.3% | 25.8% | 52.0% | 30.0% | 40.9% | 29.5% |
| o4-mini | 100.0% | 50.0% | 100.0% | 80.0% | 71.9% | 100.0% |
| GPT4.1 | 50.0% | 50.0% | 75.0% | 0.0% | 33.3% | 57.1% |

Table 3: ACPT and ICR for the baseline review agent vs. ReD. ReD lifts concerns but raises ACPTs.

| Method | ACPT-$\tau_{\text{rate}}$ | ACPT-$\tau_{0.5}$ | ICR-o3 | ICR-o4mini | ICR-GPT4.1 | ICR@M |
|---|---|---|---|---|---|---|
| Baseline Review Agent | 28.0% | 37.0% | 50.6% | 12.4% | 4.5% | 57.3% |
| ReD | 44.0% | 58.0% | 84.0% | 11.0% | 0.0% | 86.0% |

laxed metric at panel-level, *Any-of-panel ICR (ICR@M)*:

$$\text{ICR-}m = \frac{1}{N}\sum_{j=1}^{N} c_m(x_j), \qquad \text{ICR@}m = \frac{1}{N}\sum_{j=1}^{N} \bar{c}_{\text{any}}(x_j).$$

We use `GPT-5` as LLM-judge to classify whether the text feedback from review agents contains integrity-related concerns.

## 4.2 Evaluation Analysis

**Main Results.** Our main evaluation result is in Table 1. We find that acceptance is unexpectedly high under most manipulations. Single strategies already yield substantial ACPT (e.g., S1: 67.0%/82.0%; S3: 53.5%/69.7%), indicating that current review agents are easily persuaded and lack sufficient awareness to spot integrity/fabrication issues. The All strategy as a multi–setup, attains high acceptance (52.0%/69.0%), but it also maximally increases detectability (ICR@M 51.7%, o3 50.6%), suggesting that composing strategies broadens the footprint seen by detectors. Among single strategies, S1 provides the strongest acceptance with only moderate detection pressure (ICR@M 39.5%), whereas others (e.g., S3–S5) are somewhat weaker but also less detectable (ICR@M $\approx$ 30–34%). Across models, o3 is the most flag-happy (consistently higher ICR-m), while GPT-4.1 rarely flags concerns (mostly 2–8%), reinforcing that current review models have limited and uneven detection capability.

**Score distributions.** Figure 1 plots score histograms for three models across six setups (S1–S5, ALL) with the acceptance threshold marked. Overall, `o4-mini` is right-shifted—consistently placing more mass at $\geq 7$—which aligns with its higher acceptance tendency. `o3` shows larger variance and a fatter right tail (notably in S1 and ALL), producing many near-threshold and high scores; `GPT-4.1` is comparatively conservative, clustering around 6–7 with a thinner tail at 8+. Among strategies, S1 yields the strongest rightward shift for all models, while S2/S4 are milder. The ALL composition increases polarization (more mass both just below and above the threshold), explaining why it sustains high acceptance yet is easier for detectors to flag.

**Concern–Acceptance Conflict.** Conditioned on a model posting an integrity concern, we report the share that still receives an acceptance-level score in Table 2. Conflict is widespread: `o4-mini` is most contradictory (S1/S3/All: 100%; S2/S4/S5: 50–80%), `GPT-4.1` is mixed (0% in S4 but 33–75% elsewhere), and `o3` is moderate ( 26–52%). S3 (statistical theater) induces the largest cross-model conflict, and ALL further amplifies it for o4-mini (100%). These observations indicate that even agents voice concerns, yet keep acceptance-high scores, and integrity signals are not well-coupled to assessment.

## 5 Mitigation

Table 4: Evaluation results of all detectors. Across various setups, detection offers only slight gains over random. ReD is more conservative, while DetOnly is recall-oriented with higher FPR. o3 shows a positive bias, whereas GPT-4.1 tends toward negative.

| | o3 | | | | o4-mini | | | | GPT-4.1 | | | |
|---|---|---|---|---|---|---|---|---|---|---|---|---|
| Method | TPR | FPR | Acc | F1 | TPR | FPR | Acc | F1 | TPR | FPR | Acc | F1 |
| Random Guess | 50.0% | 50.0% | 50.0% | 50.0% | 50.0% | 50.0% | 50.0% | 50.0% | 50.0% | 50.0% | 50.0% | 50.0% |
| ReD | 81.6% | 44.9% | 67.0% | 72.1% | 0.0% | 8.0% | 46.0% | 0.0% | 0.0% | 0.0% | 50.0% | 0.0% |
| DetOnly | 98.0% | 84.0% | 57.0% | 69.5% | 64.0% | 74.0% | 45.0% | 53.8% | 24.0% | 12.0% | 56.0% | 35.3% |

We study two complementary interventions that make the review pipeline explicitly integrity-aware. The first augments the standard review workflow with an integrity check; the second replaces reviewing with a lightweight triage detector.

## 5.1 Setup

**Research Questions.** **RQ1 (Detection-Aware Reviewing).** How can we enhance detection awareness *within* the act of reviewing—i.e., can adding an explicit integrity-detection subtask (risk classification with cited evidence) improve sensitivity to integrity risks *while* preserving or improving core review quality (helpfulness and recommendation accuracy)?
**RQ2 (Triage without Full Reviews).** When generating a full textual review is not the first priority, can a detection-only agent reliably flag papers with potential integrity concerns at a controllable false-alarm rate and route low-risk papers to the standard pipeline?

**Mitigation Strategies.** **(i) Review-with-Detection (ReD)** We modify the review prompt to bundle a conventional rubric-based review with a mandatory detection of AI-generated content. For each document, a single model $m \in M$ outputs: (a) a standard review, (b) a binary label $l_m \in \{$HUMAN-AUTHORED, AI-GENERATED$\}$, and (c) *evidence snippets* (quoted lines) to support its finding. The model's label directly determines the final triage recommendation: a finding of AI-GENERATED results in a FLAG-FOR-AUDIT, while HUMAN-AUTHORED allows the process to PROCEED. **(ii) Detection-Only (DetOnly)** We remove the reviewing rubric and instruct the agent to perform *only* the detection of AI-generated content. In this mode, the model's sole task is to return a binary label $y_m \in \{$HUMAN-AUTHORED, AI-GENERATED$\}$ and evidence. This label serves as the direct and final triage action, without any secondary review or scoring.

**Dataset.** We construct a controlled set of 100 papers using the stratified sampling algorithm in Section 3.4: 50 are sampled from real ICLR 2025 submissions, and 50 are sampled from the generated corpus produced in Section 4.

## 5.2 Evaluation Analysis

**To RQ1: Review Outcomes.** We compare *ReD* against the initial review agent baseline (same models, same rubric, no integrity subtask). We report the review outcomes using the same metrics as in Section 4. As shown in Table 3, *ReD* increases integrity warnings overall (ICR@M $57.3\% \rightarrow 86.0\%$), driven largely by o3 ($50.6\% \rightarrow 84.0\%$), while o4-mini is roughly unchanged and GPT-4.1 collapses to $0\%$. Paradoxically, acceptance also rises substantially (ACPT$_{\tau_{\text{rate}}}$ $28.0\% \rightarrow 44.0\%$, ACPT$_{\tau_{0.5}}$ $37.0\% \rightarrow 58.0\%$). Thus, adding a detection subtask improves stated awareness but does not translate into stricter recommendations—if anything, it coexists with more accepts. This suggests the integrity signal is weakly coupled to scoring; practical deployments should gate or weight recommendations by risk rather than merely requesting detection within the review.

**To RQ2: Detection Performance.** We set three detectors on our new dataset: *Random Guess* baseline, the *ReD* integrity component, and *DetOnly*. The results are presented in Table 4. Overall, detection helps but just slightly: across models, accuracy is near the $50\%$ random baseline, with a clear lift only on **o3** (ReD $67\%$ vs. random $50\%$; DetOnly $57\%$). Comparing *ReD* and *DetOnly*, the latter is recall-seeking (higher TPR) but far noisier (much higher FPR), whereas ReD is more conservative and, on some bases, collapses (e.g., GPT-4.1 shows $0\%$ TPR for ReD). Model behavior also differs: o3 tends to judge *positive* (high flag rate; e.g., DetOnly FPR $84\%$), while GPT-4.1 tends to judge *negative* (low TPR/FPR), yielding a small accuracy gain for DetOnly ($56\%$) over random.

# 6 Conclusion and Discussion

Our findings expose a critical vulnerability: LLM review systems can be systematically deceived by presentation manipulation. Fabricated papers achieve acceptance rates up to 82%, with reviewers frequently exhibiting concern-acceptance conflicts—flagging integrity issues yet still recommending acceptance. This fundamental breakdown reveals that current AI reviewers operate more as pattern matchers than critical evaluators.

Our mitigation attempts show the inadequacy of current defenses. Detection accuracy barely exceeds random chance, and paradoxically, adding explicit integrity checks sometimes increases acceptance rates. Simply asking LLM reviewers to "be more careful" is insufficient.

The scientific community faces an urgent choice. Without immediate action to implement defense-in-depth safeguards—including provenance verification, integrity-weighted scoring, and mandatory human oversight—we risk AI-only publication loops where sophisticated fabrications overwhelm our ability to distinguish genuine research from convincing counterfeits. The integrity of scientific knowledge itself is at stake.

More detailed discussions of limitations, ethical considerations, and broader societal impacts of our work are provided in Appendix A.

## AI Agent Setup

Our work is collaborated and supported by various AI agents. For the project brainstorm and idea formalization, we use DeepResearch agent supported by `GPT-5/Gemini-2.5-Pro`. For the experiment implementation and analysis, we utilize code agents from `Copilot/Cursor` supported by `GPT-5/Claude-4-sonnet/Grok-Code-Fast-1`, which serves as the meta agent to implement and extend *BadScientist* agent framework in our research. In final stage, we rely on `Claude-4-sonnet/Gemini-2.5-Pro` for writing revision.

As stated in Section 3 and 4, our BadScientist agent framework enables various LLMs to support end-to-end paper generation, review, and post-analysis. The framework equips agents with multiple tools throughout the pipeline, including research ideation and expansion module, experiment design module, data visualization module, LaTeX compilation tool with automatic error handling to produce PDF manuscripts, and Semantic Scholar API for literature review and citation management.

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

# A Limitations, Social Impacts and Ethical Statements

## A.1 Limitations

- **Fabrication scope.** Our approach focuses on presentation manipulation rather than executable code or data artifacts. Real adversaries may employ both strategies simultaneously.
- **Model coverage.** We test three LLMs with a fixed rubric. Results may vary across model families, custom prompts, or tool-augmented reviews with code execution capabilities.
- **Generalization.** Our calibration uses a single venue and year. Cross-venue applicability, temporal drift, and adversarial adaptation remain unexplored.
- **Integrity classification.** We rely on LLM judges to extract integrity concerns from reviewer feedback, introducing potential labeling noise and bias.
- **Human oversight.** Our AI-only evaluation excludes human reviewers, program chairs, and domain experts who could mitigate observed failure modes.

## A.2 Societal Impacts

**Positive Contributions.** Our work enables three key improvements: (i) *Ecosystem hardening*—our metrics and conflict analyses provide actionable diagnostics for venues to implement integrity-aware workflows; (ii) *Enhanced transparency*—evidence-backed flagging of problematic content improves author feedback and post-hoc auditing; (iii) *Safety benchmarking*—our strategy library and evaluation framework can stress-test future review systems.

**Risks and Misuse.** We identify three primary concerns: (i) *Adversarial guidance*—our strategy descriptions could assist malicious authors, though we mitigate this by withholding exploit details and emphasizing detection; (ii) *Automation overconfidence*—venues might misinterpret modest detector improvements as justification for reduced human oversight; (iii) *Classification errors*—integrity detectors may unfairly flag legitimate work or miss sophisticated fabrications, causing reputational harm.

## A.3 Ethical Framework and Responsible Release

**Research Intent.** This work aims to strengthen scientific integrity, not facilitate fraud. Our strategy descriptions remain abstract, and we use no human subjects or clinical data.

**Data and Artifacts.** We use publicly available conference data for calibration and synthetic papers for evaluation. Our framework generates a corpus of fabricated papers and corresponding LLM reviews, which will be made available to program committees and the research community for analysis and further study of AI review vulnerabilities. Code and evaluation tools will be released with mitigation including redacted exploit details and integrity-first defaults, subject to ethical review.

**Deployment Recommendations.** We advocate for: (i) integrity-aware prompts and mandatory checklists for all AI reviewers; (ii) score-flag coupling that blocks acceptance recommendations when high-risk flags lack adequate review; (iii) provenance verification including code/data links and optional automated checks for statistics and baselines.

**Transparency Requirements.** Automated review deployments should disclose model usage and limitations to authors and committees, maintain human oversight, and log integrity evidence for auditing.

**Future Directions.** Our findings support a defense-in-depth approach combining calibrated thresholds, explicit integrity verification, automated artifact checking, and human arbitration. Future work should address executable artifact validation, structured claim verification, and cross-venue calibration to eliminate concern-acceptance conflicts and align automated review with research integrity standards.


