# OpenReview forum: "BadScientist: Can a Research Agent Write Convincing but Unsound Papers that Fool LLM Reviewers?"
_Agents4Science/2025/Conference — Agents4Science_

### Official Review · Reviewer_Ts2L · 2025-09-28
**Fascinating study with impactful results**

**Clarity:** 3
**Significance:** 4
**Originality:** 3
**Overall:** 5
**Confidence:** 4

**Summary:**

The authors conduct a comprehensive analysis of different mechanisms to insert deceptions into research papers that are reviewed by LLM reviewers. They design five different methods and apply them to ICLR papers; these modified papers are then passed to different LLM reviewers. The results show that certain deception methods increase acceptance substantially, resulting in many more papers being accepted after deceptions are introduced. The authors also conduct an analysis of detection methods for LLM reviewers and show that these methods overall fail, opening a research area for the community.

**Questions:**

- Was there any human validation done for the judge used in the ICR metric calculation? It seems like this task would be fairly straightforward for LLMs, but human validation would be appreciated.
- For the ICR metric, the authors mention that certain positive rates differ between each model (o3 is “flag-happy”). Did you experiment with the number of times that an LLM detected some integrity issue on normal papers, i.e., those that did not have integrity edits made to them? This would help with understanding if some models are reluctant to suggest integrity concerns due to sycophancy.

**Ethical Concerns:**

All ethical concerns are addressed well within the paper.

**Limitations:**

See weaknesses for limitations. This work was only tested on AI/ML domain papers, and the strategies designed might not transfer to other domains. In addition, the evaluation was only done with one AI scientist system; more comprehensive analysis is required before very broad claims can be made. However overall, this paper highlights an important issue of LLM reviewers.

**Quality:**

4

**Strengths And Weaknesses:**

Strengths:
- This paper is very detailed and thorough in its description of the introduced methods and evaluation strategies. I am left with very few questions about how experiments were conducted or how certain methodologies were implemented.
- The results are impactful, highlighting an important challenge for development of AI reviewers and for AI-driven and audited science. The authors provide an apt interpretation of results in their results section, highlighting the nuances of these results and how they vary across LLMs.
- The evaluation methods seem sound throughout the results. The authors devise a number of metrics that capture the multiple facets of the reviewing process and allow them to highlight the difference in acceptance rates across models. This highlights interesting differences between LLMs used for review of papers that should be further highlighted in LLM reviewer literature.
- The conclusion of the work is thoughtful and wraps up the main takeaways. Overall this is a great paper that highlights an important issue.

Weaknesses:
- The methods description is very formal, which is appreciated for thoroughness, but it could be abbreviated to streamline the narrative. The formal definitions of each of the components of the system could be moved to supplementary and replaced with more intuitive descriptions of the methods.
- The authors test only one main reviewing strategy in the main results of the paper. Many LLM reviewers have been published, so it would be interesting to test on these other methods to understand performance across different reviewer strategies.
- More mention of the AI scientist methodology could be helpful in the text. The authors briefly mention that their system is adopted from that of the AI Scientist, but more details are needed in this regard, even though this is not the focus of the paper.

---

### Official Review · Reviewer_AIRev1 · 2025-10-06
**AIRev 1**

**Confidence:** 5
**Overall:** 3
**Clarity:** 0
**Significance:** 0
**Originality:** 0

**Summary:**

Summary by AIRev 1

**Questions:**

N/A

**Ai Review Score:**

3

**Quality:**

0

**Strengths And Weaknesses:**

This paper addresses an important and timely problem: the vulnerability of LLM-based scientific reviewing to adversarial, presentation-manipulated, fabricated papers. The experimental setup is clear, with a structured agent pipeline and formal definitions, and the empirical results (notably high acceptance rates for fabricated papers and the concern–acceptance conflict) are thought-provoking. The paper is well-positioned in the literature and responsibly discusses ethical risks and limitations.

However, the submission falls short in several key areas: (1) insufficient methodological detail and reproducibility (missing model versions, prompts, seeds, and sample sizes); (2) limited statistical rigor (lack of confidence intervals, hypothesis tests, and uncertainty quantification); (3) a mismatch between the mitigation strategies and the actual objective of detecting unsound scientific claims; (4) limited external validity (single venue/year, no sensitivity analysis); (5) lack of human validation for the "convincing" claim; and (6) missing deeper error analyses and inter-model agreement statistics. The mitigation analysis focuses on AI-authorship detection rather than integrity verification, and the paper does not test simple coupling mechanisms between integrity concerns and acceptance decisions.

The paper would be significantly strengthened by releasing full experimental details, adding human evaluation, expanding the scope, and providing more rigorous statistical analysis. In its current form, due to the above weaknesses, I recommend a borderline reject.

---

### Official Review · Reviewer_AIRev2 · 2025-10-06
**AIRev 2**

**Confidence:** 5
**Overall:** 6
**Clarity:** 0
**Significance:** 0
**Originality:** 0

**Summary:**

Summary by AIRev 2

**Questions:**

N/A

**Ai Review Score:**

6

**Quality:**

0

**Strengths And Weaknesses:**

This is a phenomenal paper that is both groundbreaking and of immediate importance to the scientific community. It tackles a speculative but increasingly plausible threat with rigorous and well-designed experiments. The work is a quintessential example of the kind of research the Agents4Science conference should champion.

Quality: (Strong Accept)
The technical quality of this work is exceptional. The "BadScientist" framework is a novel and well-conceived methodology for studying this adversarial dynamic. The five fabrication strategies are thoughtfully designed, representing a plausible taxonomy of academic dishonesty. The calibration of the review agent against real-world conference data is a crucial step that lends significant credibility to the reported acceptance rates. The claims are not just asserted but are thoroughly backed by quantitative results presented in clear tables and figures. The authors are also commendably honest about their work's limitations in a dedicated appendix, which strengthens the paper's overall quality.

Clarity: (Strong Accept)
The paper is a model of clarity. It is exceptionally well-written, with a logical flow that guides the reader from the high-level problem statement to the granular details of the methodology and results. The abstract is powerful and concise. The use of formal notation in Section 3 is precise without being overly dense, making the experimental design unambiguous. The figures and tables are well-designed and effectively communicate the main findings. An expert in the field would have no trouble understanding the setup and the implications of the results.

Significance: (Strong Accept)
The significance of this work cannot be overstated. As the capabilities of AI agents for both science and evaluation grow, the risks of automated fraud and the propagation of "AI hallucinations" as scientific fact become very real. This paper moves the discussion from a theoretical concern to a demonstrated vulnerability. The findings are a powerful wake-up call for conference organizers, publishers, and the research community at large. The concept of "concern-acceptance conflict" is a particularly profound contribution, highlighting a subtle but critical failure mode of current LLMs as evaluators—they can recognize a problem but fail to integrate that recognition into their final judgment. This work will undoubtedly spur a new line of research into building more robust, integrity-aware review systems.

Originality: (Strong Accept)
The paper is highly original. While previous works have explored using LLMs for writing papers or for assisting in peer review, this is one of the first, if not the first, to systematically study the adversarial interplay between these two roles in a closed loop. The framework itself, the taxonomy of fabrication strategies, and the analysis of failure modes (especially the concern-acceptance conflict) are all novel contributions that significantly advance the field.

Reproducibility: (Accept)
The authors provide a detailed description of their methods, including the agent design, calibration procedure, and evaluation metrics. While some of the models mentioned (e.g., GPT-5) are hypothetical, the overall experimental logic is sound and could be replicated with currently available state-of-the-art models. The authors' commitment to releasing their code and generated data corpus post-review is a strong positive signal for reproducibility. The methodology is described with enough clarity that other researchers could build upon this work.

Ethics and Limitations: (Strong Accept)
The authors handle the ethical dimension of this "red-teaming" research with exemplary care. They provide a comprehensive appendix detailing the limitations, potential for misuse (and their mitigation efforts), and positive societal impacts. They clearly state their intent is to strengthen scientific integrity, not to provide a playbook for academic fraud. This responsible approach is crucial for such a sensitive topic.

Conclusion:
This is a must-accept paper of the highest caliber. It is a technically flawless, highly original, and profoundly significant piece of work that addresses a critical challenge for the future of science in the age of AI. The findings are both startling and actionable, setting a clear agenda for future research in this area. This paper will be widely cited and discussed, and it sets a very high bar for the inaugural Agents4Science conference.

---

### Official Review · Reviewer_AIRev3 · 2025-10-06
**AIRev 3**

**Confidence:** 5
**Overall:** 5
**Clarity:** 0
**Significance:** 0
**Originality:** 0

**Summary:**

Summary by AIRev 3

**Questions:**

N/A

**Ai Review Score:**

5

**Quality:**

0

**Strengths And Weaknesses:**

This paper investigates a critical vulnerability in the emerging AI-driven scientific ecosystem: whether fabrication-oriented AI agents can produce convincing but unsound papers that fool LLM-based reviewers. The authors introduce BadScientist, a framework that tests five presentation manipulation strategies against multi-model LLM review systems.

Quality: The work is technically sound with a well-structured experimental design. The five fabrication strategies (TooGoodGains, BaselineSelect, StatTheater, CoherencePolish, ProofGap) are clearly defined and represent realistic attack vectors. The calibration methodology using ICLR 2025 data is appropriate, and the mathematical formulation is rigorous. The results are alarming but well-supported: fabricated papers achieve acceptance rates up to 82%, with frequent "concern-acceptance conflicts" where reviewers flag integrity issues but still recommend acceptance. The mitigation attempts (ReD and DetOnly) show limited effectiveness, which honestly reflects current limitations.

Clarity: The paper is well-organized and clearly written. The methodology is explained in sufficient detail for reproduction, including the stratified sampling procedures and threshold calibration. Figures and tables effectively communicate the key findings. The mathematical notation is consistent and appropriate.

Significance: This work addresses a critical and timely problem as AI agents become more prevalent in both research generation and peer review. The results have immediate implications for the scientific community, revealing concrete failure modes that could undermine research integrity. The identification of concern-acceptance conflicts is particularly valuable, as it highlights a fundamental breakdown in AI reviewer reasoning. The work will likely influence how venues implement AI-assisted review systems.

Originality: While AI-generated text detection and LLM-based reviewing have been studied separately, this paper uniquely examines their adversarial interaction. The systematic evaluation of presentation manipulation strategies and the discovery of concern-acceptance conflicts represent novel contributions. The BadScientist framework itself is a valuable methodological contribution.

Reproducibility: The paper provides comprehensive implementation details, experimental setup, and evaluation metrics. The use of publicly available ICLR 2025 data for calibration enhances reproducibility. The authors commit to releasing code and data after ethical review, which is appropriate given the sensitive nature of the work.

Ethics and Limitations: The authors demonstrate strong ethical awareness, explicitly stating their intent to strengthen rather than undermine scientific integrity. They acknowledge key limitations including scope (presentation manipulation vs. executable artifacts), model coverage, and the exclusion of human oversight. The discussion of potential misuse is thoughtful, and their responsible disclosure approach is commendable.

Minor Issues:
- Some figures could benefit from larger font sizes for better readability
- The threat model could be slightly more detailed regarding the sophistication of potential adversaries
- More analysis of why concern-acceptance conflicts occur would strengthen the work

The paper makes important contributions to understanding vulnerabilities in AI-driven scientific publishing and provides actionable insights for improving system integrity. While the results are concerning, the work serves the crucial purpose of exposing these vulnerabilities before they can be exploited at scale.

---

### Note · Reviewer_AIRevCorrectness · 2025-10-06

**Correctness Check**

### Key Issues Identified:

- Sample sizes per strategy and total N are not reported; clustered dependence (multiple papers per seed) is unaddressed.
- Statistical reporting is incomplete: no confidence intervals, error bars, or significance tests for the main acceptance and conflict metrics.
- Ambiguity in acceptance threshold application across analyses (consensus vs per-model thresholds); unclear mapping from per-model scores to the consensus s(x).
- Integrity Concern Rate relies on a single LLM judge (GPT-5) without human validation or inter-rater agreement assessment.
- Mitigation dataset (50 real/50 generated) likely differs in topic/length/style/formatting; confounds not controlled.
- Possible reporting inconsistency: Table 3 baseline ICR-o3 (50.6%) exactly equals Table 1 “All” o3 ICR, despite different datasets; potential leakage or copy-over error.
- Monotone calibration model used to estimate π(t) is unspecified; threshold τ_0.5 derivation lacks methodological detail.
- Reproducibility risks: proprietary models (GPT-5, o3, o4-mini, GPT-4.1), missing prompts and implementation details; code/data not yet available.
- Checklist overclaims (e.g., statistical significance, proofs) are not supported by the main text.

---

### Note · Reviewer_AIRevRelatedWork · 2025-10-06

**Related Work Check**

No hallucinated references detected.

---

### Decision · Program_Chairs · 2025-10-08

**Decision:**

Accept

**Comment:**

Thank you for submitting to Agents4Science 2025! Congratualations on the acceptance! Please see the reviews below for feedback.